# Dietary diversity, undernutrition, and predictors among pregnant adolescents and young women attending Gulu University teaching hospitals in northern Uganda

Emmanuel Musinguzi[1], Peninah Nannono[1], Moreen Ampumuza[1], Mathew Kilomero[1], Brenda Nakitto[1], Yakobo Nsubuga[1], Byron Awekonimungu[1], Rebecca Apio[1], Moses Komakech[1], Luke Odongo[1], Pebalo Francis Pebolo[1], Felix Bongomin[1,2]*

**1** Faculty of Medicine, Gulu University, Gulu, Uganda, **2** Department of Internal Medicine, Gulu Regional Referral Hospital, Gulu, Uganda

* drbongomin@gmail.com

## Abstract

### Background

Undernutrition has deleterious consequences to both the mother and the unborn child, significantly contributing to maternal and neonatal morbidity and mortality. We assessed dietary diversity, the prevalence, and predictors of undernutrition among pregnant adolescents and young women (PAYW) attending antenatal clinic (ANC) at two large teaching hospitals in northern Uganda.

### Methods

Between 12th June 2023 to 27th October 2023, we conducted a facility-based, cross-sectional study at Gulu Regional Referral Hospital (GRRH) and St Mary's Hospital Lacor (SMHL), both in Gulu district, Uganda. We recruited PAYW aged 15–24 years attending ANCs. Sociodemographic and clinical characteristics of the study participants were collected using a pre-tested, semi-structured questionnaire. Undernutrition was defined as a mid-upper arm circumference (MUAC) of < 23 cm. Modified Poisson regression analysis was performed to determine factors independently associated with undernutrition. Data analyses was performed using STATA version 17.0. A p<0.05 was considered statistically significant.

### Results

A total of 324 participants, with a mean age of 21.2±2.2 years were enrolled. About 62.0% (n = 201) of the participants dewormed during pregnancy. The prevalence of undernutrition was 12.7% [n = 41]. Prevalence was higher among participants who maintained pre-pregnancy diet (adjusted prevalence ratio [aPR] = 2.27, 95% Confidence Interval [CI]: 1.26–4.05, p = 0.006), those who did not receive nutritional education (aPR = 2.25, 95% CI: 1.21–4.20, p = 0.011) and consumption of non-green leafy vegetables (aPR = 4.62

**Data Availability Statement:** All relevant data are within the manuscript and its Supporting Information files.

**Funding:** The author(s) received no specific funding for this work.

**Competing interests:** The authors have declared that no competing interests exist.

**Abbreviations:** ANC, Antenatal care; AYW, Adolescent and Young Women; GRRH, Gulu Regional Referral Hospital; GWG, Gestational Weight Gain; MUAC, Mid upper arm circumference; OR, Odds Ratio; PAYW, Pregnant Adolescent and Young Women; SDG, Sustainable Developmental Goals; SMHL, St Mary's Hospital Lacor; USAID, United States Agency for International Development; WHO, World Health Organization.

95% CI: 1.64–13.01, p = 0.004). The prevalence of undernutrition was lower among participants who consumed milk and milk products (aPR = 0.44 95% CI: 0.24–0.81, p = 0.009) and among those who consumed fish and seafood compared to those who did not (aPR = 0.45 95% CI: 0.20–1.00, p = 0.050).

## Conclusions

About 1 in 8 of PAYW attending GRRH or SMHL had undernutrition, particularly those who lacked education about feeding habits during pregnancy and limited access to milk and milk products, fish and seafoods. We recommend health workers to offer timely education of pregnant adolescent and young women mothers about good feeding habits during pregnancy, appropriate monitoring of weight gain and physiological changes during pregnancy.

## Background

A safe and healthy diet contributes to optimal state of health and protects against all forms of malnutrition. However unhealthy diet is one of the top leading causes of the increased burden of disease [1]. Yet an estimated two billion people are at risk of vitamin A, iodine, and iron deficiency globally, majority being among pregnant women and young children in the Southeast Asia and sub-Saharan Africa [2].

Nutrient requirements including those of for energy, protein, iron, calcium, and others increase in adolescence to support adequate growth and development. In settings where dietary intakes are suboptimal, anemia and micronutrient deficiencies are high. Growth velocity increases during puberty when the peak height velocity occurs and catch up is possible; in girls, about 15–25% of adult height is attained. A premature pregnancy can halt the linear growth and increase the risk of adverse birth outcome [3].

Systematic review and meta-analysis about the burden and the determinants of Malnutrition among pregnant women in Africa, revealed; that using a random effect model, the overall pooled prevalence of malnutrition among pregnant women was 23.5%, with rural residency, low educational status of partners, multiple pregnancy and poor nutritional indicators positively determining malnutrition. On the contrary better household economic status negatively determined malnutrition. Therefore, a significant number of the pregnant population in Africa is suffering from malnutrition [4].

Adolescent under nutrition is a wide spread problem prominently in economically developing countries including Uganda [5]. The prevalence of adolescent under nutrition is still high and on the rise. For example, according to the 2011 and 2016 Uganda demographic and health survey reports, the prevalence of thinness in adolescent girls (15–19 years) was 13% and 14%, respectively [6].

Pregnant women, especially adolescents are at increased risk of under nutrition and studies in different areas have shown that under nutrition is significantly affected by various factors. However, statistics in Gulu are not known; therefore, this study was aimed at determining the prevalence and predictors with undernutrition among pregnant adolescents and young women (PAYW) attending antenatal care (ANC) at Gulu Regional Referral Hospital (GRRH) and St. Mary's Hospital- Lacor (SMHL), Uganda.

## Methods

### Study design

This was a facility-based, observational, cross-sectional study with a quantitative approach. Data was collected between 12[th] June 2023 to 27[th] October 2023.

### Study setting

The study was carried out at SMHL and GRRH. In both hospitals, the average attendance of ANC clinic was about 10 to 20 PAYWs per day. Both ANC clinics ran from Monday to Friday. The attendees are mainly people from the Northern region of Uganda, especially Gulu and neighboring districts. Both clinics have resident midwives and gynecologists, with additional work force being derived from a pool of nutritionists, and medical officers.

### Study population

We recruited PAYW attending ANC at GRRH and SMHL registered with ANC at GRRH or SMHL aged 15–24 years who provided written informed consent. Participants with missing ANC data were excluded.

### Sample size estimation

We calculated a sample size of 329 using the Kish Leslie formula, with the prevalence of under-nutrition of 26.4% [7], type 1 error of 5% (1.96) and 95% confidence, assuming 10% attrition rate at a precision of 5%.

### Study variables

The dependent variable was undernutrition defined by MUAC<23.0cm. The independent variables were socio-demographic factors: age, marital status, husband education maternal education, family size, polygamy. Maternal related factor: - parity, family planning utilization before current pregnancy, birth interval, receiving iron supplementation, ANC follows up, ANC satisfaction, nutrition knowledge, Illness, History of abortion, history of stillbirth, dietary diversification (24-h recall), meal frequency, Socio-cultural factors: -food taboo and food restriction during pregnancy, decision making on household assets, family stable food. Economic factors: -Family source of food, farmland ownership, employment (maternal& husband job) status, household income, wealth index. Hygiene and sanitation-related factors: -access to water and sanitation facilities, such as latrine availability & utilization, family source of water, distance to get water. Food Security-related factors - food accessibility and availability

### Data collection procedure

An interviewer-administered structured questionnaire was used to collect the data. The structured questionnaire was prepared in English. The questionnaire was translated to *leb Acholi* language verbally to those who didn't understand English. The tool included socio-demographic factors, Maternal related factor, Socio-cultural factors, Economic factors, Hygiene and Sanitation related factors, and food Security related f. Data collection was performed by the investigators and 2 trained nurses at each study sites 5 days a week. To ensure the quality of the data, properly designed data collection instrument and training of data collectors and supervisors was done, the data collectors and the supervisor were given training for three days on procedures, techniques, ways of collecting the data, and monitoring the procedure. The collected data was reviewed and checked for completeness at the end of each week.

## Study measurement

**Mid-upper arm circumference (MUAC).** To determine nutrition status of pregnant women MUAC was measured by considering the mothers in Frankfurt plane and sideways to measure the left side, arms hanging loosely at the side with the palm facing inward, taken at marked midpoint of upper left arm. To identify the midpoint for the measurements, a tap measure is placed between two points i.e. (olecranon process and the acromion), half distance between these two points is considered the point for measuring MUAC. MUAC< 23 cm as used was considered as under nutrition in this study. MUAC is a simple and widely used anthropometric measurement to assess nutritional status, particularly in children and pregnant women. It involves measuring the circumference of the upper arm at the midpoint between the shoulder and the elbow. MUAC is a reliable indicator of acute malnutrition because it reflects changes in muscle and fat mass. In pregnant women, MUAC can help identify those at risk of malnutrition and related complications. The ease of use and minimal equipment required make MUAC an essential tool in community and clinical settings, especially in low-resource environments.

**Dietary diversity score (DDS).** Dietary diversity information of individual respondents was collected using the 24-hour recall method and women dietary diversity score model questionnaire of nine food groups with food listing method in which list of food items replaced by common foods in local context was included in the questionnaire. The dietary diversity score (DDS) is an important indicator of dietary quality and nutritional adequacy. A higher DDS suggests a more varied diet, which is associated with better nutrient intake and overall health outcomes. The DDS is calculated by counting the number of different food groups consumed, with common groups including cereals, vegetables, fruits, meat, dairy, legumes, nuts, and oils. This measure is particularly useful in evaluating the effectiveness of nutrition interventions and identifying populations at risk of dietary deficiencies. The simplicity of DDS makes it a practical tool for use in both research and programmatic settings to monitor and improve dietary patterns.

## Data management and quality control

The questionnaires were thoroughly checked by the research assistants before submission to make sure that they had been properly filled and all questions answered. The data was kept securely with a password and the consent forms were kept under lock and key. Data was coded and exported for analysis using STATA version 17. Prior to the data collection, the research assistants were trained. The instrument was pretested in 5% of the sample size. Pretest was conducted on individuals with similar characteristics of the study population who were not a part of the actual study. Based on the pretested results, the instrument was modified and changed.

## Data analysis

Data was cleaned and analyzed using STATA version 17.0 (STATA, College Station, Texas, USA). Frequencies were used at univariate level. Continuous variables were described using mean and standard deviation, categorical variables were described using frequencies and percentages. The results were displayed using tables and graphs.

At bivariate level, inferential statistic was done using the chi-square test for proportion to analyze the association between independent variables and undernutrition. Factors with p<0.2 were included for multivariate analysis. With prevalence of undernutrition >10%, we used modified Poisson regression analysis at multivariate level to determine factors that were independently associated with undernutrition. Factors with p<0.05 were considered

significantly associated with undernutrition. Factors that were statistically significant were interpreted based on adjusted prevalece ratio (aPR). Prevalence ratio (PR), 95% confidence interval (CI) and p-values were reported. We accounted for all important confounders such as age, household income, and marital status in the multivariable model. Missing data were addressed using multiple imputation techniques to ensure that the results were robust and to minimize potential biases due to incomplete information.

## Ethical considerations

Ethical approval was sought from Gulu University Research and Ethics Committee (GUREC; approval 2022–435). All study participants were provided written informed consent before the interview could be conducted and the contents of the form were thoroughly explained to the participants before appending their signatures. Their participation in the study was entirely voluntary. The participants were given the freedom to ask questions before, during and after the interview. To maintain anonymity, participants' names were not used on the questionnaires, only codes were used. Confidentiality was ensured throughout the study and privacy provided during the interview session. The participants were not subjected to any physical harm, and they were free to withdraw from the study at any time before analysis was done. All the information obtained from this study was strictly for academic purposes and not for any other reasons against the respondents.

## Results

### Study enrollment

Of 329 potential participants screened, 324 were eligible and were enrolled into the study, given a response rate of 98%. **Fig 1**

### Socio-demographic characteristics of participants

A total of 324 participants, with a mean age of 21.2±2.2 years were enrolled. Most participants were in the age range 19–22 years (50.6%, n = 164). About 48.1% (n = 156) of the participants had completed primary education, (59.3%, n = 192) were nulliparous. Most respondents

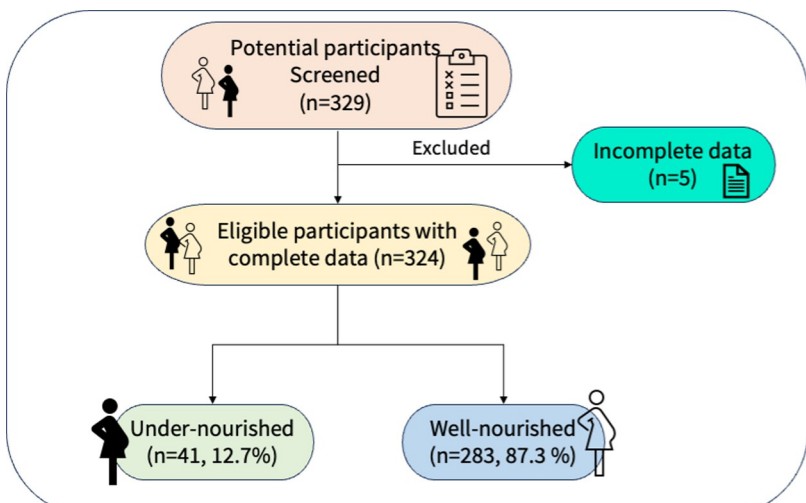

**Fig 1. Study flow diagram.**

reported antenatal attendance, with 62.0% (n = 201) having undergone deworming during pregnancy, **Table 1**.

## Prevalence of undernutrition among pregnant adolescents and young women in Gulu regional referral hospital and St Mary's Hospital Lacor

The mean MAUC of all participants was 25.5± 2.8 centimeters. The prevalence of undernutrition was 12.7% (n = 41), **Fig 2**. The mean Dietary Diversity Score is 12.7±1.6, with 87.3% (n = 283) of the participants having an adequate dietary diversity score, while 12.7% (n = 41) have a minimally inadequate score, **Table 2**.

## Individual characteristics of pregnant adolescents and young women in Gulu regional referral hospital and St Mary's Hospital Lacor

Among the total participants, 48.8% of those reported changes experiencing no undernutrition, Family planning for child spacing was practiced by 38.1% (n = 123) of respondents, with 31.7% (n = 13) facing undernutrition. The mean time taken before delivering another child was 1.670 (standard deviation 1.820), **Table 3**.

## Health sector related characteristics of pregnant adolescents and young women in Gulu regional referral hospital and St Mary's Hospital Lacor

The mean time taken to reach the hospital was 41.784 (standard deviation 39.235) minutes by walk for all participants. Regarding assessments on feeding during hospital visits, 73.2% (n = 30) were facing undernutrition, **Table 4**.

## 24-Hour recall of type of food taken at different times

For the overall distribution of occasions, most meals were reported as breakfast (40.4%), followed by lunch (28.8%), late-night meals (19.7%), dinner (9.2%), and brunch (2.0%). Notably, tea was the most frequently consumed item, with 53.5% during breakfast, 8.2% during dinner, and 14.9% during late-night meals. Posho was consumed significantly during lunch (66.2%) and dinner (46.6%), **Table 5**.

## Women dietary diversity question model of pregnant adolescents and young women in Gulu regional referral hospital And St Mary's Hospital Lacor

Among the food categories, white root tubers and plantains, dark green leafy vegetables were consumed most at 99.7%, (n = 323), followed by foods made from grains at 99.4% (n = 322), pulses at 99.1% (n = 321), nuts and seeds at 97.5% (n = 316) and the least consumed was organ meat at 72.2% (n = 234), **Table 6**.

## Factors associated with undernutrition among pregnant adolescents and young women in Gulu regional referral hospital and St Mary's Hospital Lacor

Factors statistically significantly associated with higher prevalence of undernutrition were not changing from the time they became pregnant (aPR: 2.27, 95% CI: 1.26–4.05, p = 0.006), not being taught about feeding habits during pregnancy (aPR: 2.25, 95% CI: 1.21–4.20, p = 0.011). Consumption of milk and milk products (aPR: 0.44, 95% CI: 0.24–0.81, p = 0.009), fish and seafood consumption (aPR: 0.45, 95% CI: 0.20–1.00, p = 0.050) and the consumption of other

**Table 1. Socio-demographic characteristics of pregnant adolescents and young women in Gulu regional referral hospital and St Mary's Hospital Lacor.**

| Variable | Freq (%) |
|---|---|
| **Health facility** | |
| St Mary's Hospital Lacor | 211 (65.1%) |
| Gulu Regional Referral Hospital | 113 (34.9%) |
| **Age Mean (Standard Deviation)** | 21.2 (2.2) |
| 15–18 | 42 (13.0%) |
| 19–22 | 164 (50.6%) |
| 23–26 | 118 (36.4%) |
| **Education level** | |
| No formal education | 6 (1.9%) |
| Primary | 156 (48.1%) |
| Secondary | 139 (42.9%) |
| Post-secondary | 23 (7.1%) |
| **Obstetrics and gynecological history** | |
| **Parity** | |
| 0 | 192 (59.3%) |
| 1 | 102 (31.5%) |
| 2 | 24 (7.4%) |
| 3 | 4 (1.2%) |
| 4 | 2 (0.6%) |
| **Gravidity** | |
| 1 | 164 (50.9%) |
| 2 | 111 (34.3%) |
| 3 | 34 (10.5%) |
| 4 | 11 (3.4%) |
| 5 | 3 (0.9%) |
| **Abortions** | |
| 0 | 273 (85.0%) |
| 1 | 40 (12.5%) |
| 2 | 4 (1.2%) |
| 3 | 2 (0.6%) |
| 4 | 1 (0.3%) |
| 8 | 1 (0.3%) |
| **Antenatal attendance, frequency** | |
| 1 | 68 (21.0%) |
| 2 | 41 (12.7%) |
| 3 | 83 (25.6%) |
| 4 | 62 (19.1%) |
| 5 | 36 (11.1%) |
| 6 | 17 (5.2%) |
| 7 | 12 (3.7%) |
| 8 | 4 (1.2%) |
| 9 | 1 (0.3%) |
| **Last deworming** | |
| **Yes** | 201 (62.0%) |
| **No** | 123 (38.0%) |
| **Location of current place of residence** | |

*(Continued)*

**Table 1.** (Continued)

| Variable | Freq (%) |
|---|---|
| **Urban** | 194 (59.9%) |
| **Rural** | 130 (40.1%) |
| **Marital status** | |
| **Married** | 212 (65.4%) |
| **Unmarried** | 112 (34.6%) |
| **Estimate household income** | 200,000 (100,000–350,000) |
| **Estimate Monthly household income, Ugandan shillings (UGX); $1 = 3,750UGX** | |
| **0–500,000** | 289 (89.8%) |
| **>500,000** | 33 (10.2%) |

vegetables (aPR:4.62, 95% CI: 1.64–13.01, p = 0.004) were protective against undernutrition, Table 7.

## Discussion

In this study, we aimed to assess the prevalence and factors associated with undernutrition among PAYW attending tertiary teaching hospitals in northern Uganda. The overall prevalence of undernutrition was 12.7%, consistent with previous studies [8]. A large proportion (51.2%) of the respondents had not changed their feeding from the time they got pregnant, and these contributed to majority (68.3%) of all the undernourished mothers in the study. Furthermore our study revealed that 15.9% of the participants were not taught about feeding habits during the ANC visits, and this is in line with a finding in a Addis Ababa [9]

This study also showed that 12.7% of the participants were having a minimally inadequate diet; these contributed a significant proportion of the undernourished mothers. Specifically; 36.6% of the participants not taking milk and milk products, 17.1% of the participants not consuming meat and poultry, 19.5% not eating fish and sea food and 9.8% of the participants not eating vitamin A rich fruits as previously described [10].

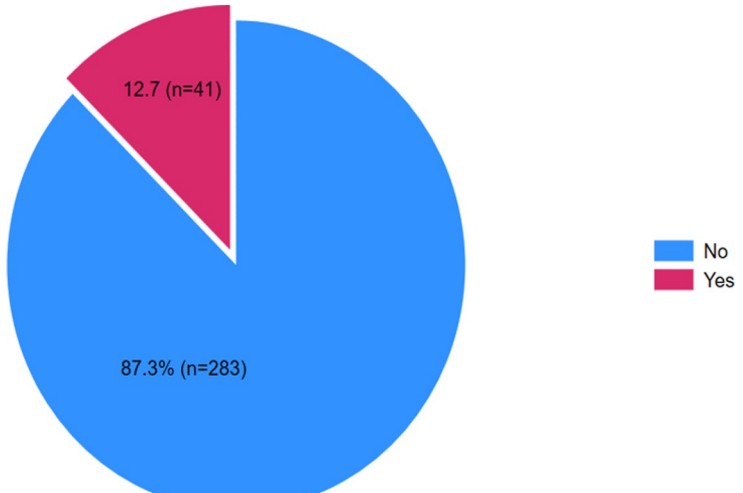

**Fig 2. Prevalence of undernutrition among pregnant adolescents and young women.**

**Table 2. Prevalence of undernutrition among pregnant adolescents and young women in Gulu regional referral hospital and St Mary's Hospital Lacor.**

| Variable | Freq (%) |
|---|---|
| MAUC. Mean (standard deviation) | 25.5 (2.8) |
| Undernutrition | |
| No | 283 (87.3%) |
| Yes | 41 (12.7%) |
| Dietary Diversity score. Mean (standard deviation) | 12.7 (1.6) |
| Minimally Inadequate | 41 (12.7%) |
| Adequate | 283 (87.3%) |

The high prevalence of undernutrition in our study doubles findings in other parts of Uganda in which level of under nutrition defined as underweight 6.9%[8]. The differences could be related to the study settings, Northern Uganda being a region recovering from over 20 years civil wars and has the one of the highest poverty indices in the country, but also due to the methods of measurements of undernutrition and the population studied. Similar prevalence was found other studies such as those done in neighboring Sudan (12.5%). Conversely, the prevalence of our study is higher compared to studies conducted in Lamwo district (8.6%) of Uganda, 11% in Tanzania[11], Northeast India(10.5%)[12], Brazil(5.8%) [13]. The global prevalence of undernutrition in Africa is 23.5%. [14] The prevalence of undernutrition in our study was smaller compared to the study done in Gindeberet district Ethiopia(17.7%) [15], Gondar (14.4%) [16], Konso district Southern Ethiopia(43.1%) [17], Western Ethiopia (39.2%) [18], Kenya(27%) [19], Aletawondo, Southern region of Ethiopia which is 15.1% [20]and findings from a study conducted in South Africa (13%),[21].

Having changed (change in pre-pregnancy diet, weight, and physiological changes) from the time one gets pregnant was one of the determinants, which were independently associated with undernutrition among pregnant adolescents and young women. This study revealed that 48.8% of the participants had changedwhich correlates to [22] where women lost some control over their lives especially in the first trimester and 25.5% developed depressive symptoms during the 3<sup>rd</sup> trimester which was a result of inadequate money to afford food and proper shelter and lack of exercise which lead to poor health status during pregnancy [23]

**Table 3. Individual characteristics of pregnant adolescents and young women in Gulu regional referral hospital and St Mary's Hospital Lacor.**

| Variable | Undernutrition | | | |
|---|---|---|---|---|
| | No | Yes | ALL | p-value |
| | 283 (87.3%) | 41 (12.7%) | 324 (100.0%) | |
| Have you changed from the time you became pregnant? * | | | | |
| Yes | 137 (48.8%) | 13 (31.7%) | 150 (46.6%) | 0.041 |
| No | 144 (51.2%) | 28 (68.3%) | 172 (53.4%) | |
| Do you use family planning to space your children? | | | | |
| Yes | 110 (39.0%) | 13 (31.7%) | 123 (38.1%) | 0.368 |
| No | 172 (61.0%) | 28 (68.3%) | 200 (61.9%) | |
| Have you been receiving iron(brown) tablets during hospital? | | | | |
| Yes | 215 (76.2%) | 29 (70.7%) | 244 (75.5%) | 0.443 |
| No | 67 (23.8%) | 12 (29.3%) | 79 (24.5%) | |
| How long do you take before you deliver another child? Mean (standard deviation) | 1.693 (1.840) | 1.512 (1.690) | 1.670 (1.820) | 0.554 |

* = The changes include the change in pre-pregnancy diet, weight and physiological changes

**Table 4. Health sector related characteristics of pregnant adolescents and young women in Gulu regional referral hospital and St Mary's Hospital Lacor.**

| Variable | Undernutrition | | | |
|---|---|---|---|---|
| | No | Yes | ALL | p-value |
| | 283 (87.3%) | 41 (12.7%) | 324 (100.0%) | |
| How long do you take to reach the hospital? Mean (standard deviation), minutes | 41.442 (39.522) | 44.146 (37.582) | 41.784 (39.235) | 0.681 |
| Are you always assessed about your feeding during the visits in this hospital | | | | |
| Yes | 226 (80.1%) | 30 (73.2%) | 256 (79.3%) | 0.304 |
| No | 56 (19.9%) | 11 (26.8%) | 67 (20.7%) | |
| have you ever been taught about feeding habits during pregnancy? | | | | |
| Yes | 238 (84.1%) | 30 (73.2%) | 268 (82.7%) | 0.084 |
| No | 45 (15.9%) | 11 (26.8%) | 56 (17.3%) | |

About 15.9% of the participants were not taught of the feeding habits during the ANC visits this could have caused knowledge gap about the required dietary modifications during pregnancy this in line with [9] that illustrated increased knowledge of pregnant women on nutrition by 43.1% after provision of nutritional health talks during ANC and improved dietary modifications by 36.9% and according to [24] nutritional education and counselling during ANC greatly improved maternal diets including practices and consumption of specific micro and macronutrients. In a paper by [25] it was stated that in low resource settings like Uganda the most health promoting intervention towards nutrition was nutrition education.

With mean dietary diversity score of 12.651, 12.7%(n = 41) of the participants were having a minimally inadequate diet with 36.6%(n = 15) of the participants not taking milk and milk products, 17.1% (n = 7) of the participants not consuming meat and poultry, 19.5%(n = 8) not eating fish and sea food and 9.8%(n = 4) of the participants not eating vitamin A rich fruits. A study done in Kayunga revealed that more than 80% of women did not achieve the minimum dietary diversity and less than 50% consumed foods rich in plant and animal proteins and 80% of their diet lacked vitamin A-rich fruits, vegetables and dark green leafy vegetables.[10]. According to [26] most of the women in Northern Uganda consume food that is

**Table 5. 24-Hour recall of type of food taken at different times.**

| Item | Occasion | | | | |
|---|---|---|---|---|---|
| | Breakfast 390 (40.4%) | Brunch 19 (2.0%) | Lunch 278 (28.8%) | Dinner 89 (9.2%) | Late night meal 190 (19.7%) |
| Bread | 5 (1.7%) | 2 (11.1%) | 7 (3.3%) | 0 (0.0%) | 2 (1.5%) |
| Cassava | 14 (4.7%) | 2 (11.1%) | 21 (10.0%) | 7 (9.6%) | 10 (7.5%) |
| Chapati | 6 (2.0%) | 0 (0.0%) | 0 (0.0%) | 0 (0.0%) | 1 (0.7%) |
| Irish | 8 (2.7%) | 0 (0.0%) | 3 (1.4%) | 3 (4.1%) | 0 (0.0%) |
| Juice | 3 (1.0%) | 1 (5.6%) | 1 (0.5%) | 2 (2.7%) | 2 (1.5%) |
| Maize | 4 (1.3%) | 0 (0.0%) | 0 (0.0%) | 0 (0.0%) | 0 (0.0%) |
| Matooke | 6 (2.0%) | 0 (0.0%) | 4 (1.9%) | 2 (2.7%) | 2 (1.5%) |
| Tea | 160 (53.5%) | 0 (0.0%) | 1 (0.5%) | 6 (8.2%) | 20 (14.9%) |
| Porridge | 31 (10.4%) | 0 (0.0%) | 3 (1.4%) | 4 (5.5%) | 3 (2.2%) |
| Posho | 43 (14.4%) | 8 (44.4%) | 139 (66.2%) | 34 (46.6%) | 65 (48.5%) |
| Rice | 9 (3.0%) | 3 (16.7%) | 27 (12.9%) | 8 (11.0%) | 23 (17.2%) |
| Simsim | 0 (0.0%) | 0 (0.0%) | 1 (0.5%) | 0 (0.0%) | 1 (0.7%) |
| Spaghetti | 1 (0.3%) | 0 (0.0%) | 1 (0.5%) | 1 (1.4%) | 0 (0.0%) |
| Sugarcane | 0 (0.0%) | 0 (0.0%) | 0 (0.0%) | 1 (1.4%) | 1 (0.7%) |
| Sweet potato | 9 (3.0%) | 2 (11.1%) | 2 (1.0%) | 5 (6.8%) | 4 (3.0%) |

**Table 6.  Women dietary diversity question model of pregnant adolescents and young women in Gulu regional referral hospital and St Mary's Hospital Lacor.**

| Food category | Undernutrition | | | p-value |
|---|---|---|---|---|
| | No | Yes | ALL | |
| | 283 (87.3%) | 41 (12.7%) | 324 (100.0%) | |
| **Foods made from grains** | | | | |
| No | 2 (0.7%) | 0 (0.0%) | 2 (0.6%) | 0.589 |
| Yes | 281 (99.3%) | 41 (100.0%) | 322 (99.4%) | |
| **White roots and tubers and plantains** | | | | |
| No | 1 (0.4%) | 0 (0.0%) | 1 (0.3%) | 0.703 |
| Yes | 282 (99.6%) | 41 (100.0%) | 323 (99.7%) | |
| **Pulses (beans, peas and lentils)** | | | | |
| No | 3 (1.1%) | 0 (0.0%) | 3 (0.9%) | 0.508 |
| Yes | 280 (98.9%) | 41 (100.0%) | 321 (99.1%) | |
| **Nuts and seeds** | | | | |
| No | 7 (2.5%) | 1 (2.4%) | 8 (2.5%) | 0.989 |
| Yes | 276 (97.5%) | 40 (97.6%) | 316 (97.5%) | |
| **Milk and milk product** | | | | |
| No | 55 (19.4%) | 15 (36.6%) | 70 (21.6%) | 0.013 |
| Yes | 228 (80.6%) | 26 (63.4%) | 254 (78.4%) | |
| **Organ meat** | | | | |
| No | 79 (27.9%) | 11 (26.8%) | 90 (27.8%) | 0.885 |
| Yes | 204 (72.1%) | 30 (73.2%) | 234 (72.2%) | |
| **Meat and poultry** | | | | |
| No | 21 (7.4%) | 7 (17.1%) | 28 (8.6%) | 0.040 |
| Yes | 262 (92.6%) | 34 (82.9%) | 296 (91.4%) | |
| **Fish and seafood** | | | | |
| No | 19 (6.7%) | 8 (19.5%) | 27 (8.3%) | 0.006 |
| Yes | 264 (93.3%) | 33 (80.5%) | 297 (91.7%) | |
| **Eggs** | | | | |
| No | 33 (11.7%) | 8 (19.5%) | 41 (12.7%) | 0.158 |
| Yes | 250 (88.3%) | 33 (80.5%) | 283 (87.3%) | |
| **Vitamin A-rich vegetables, roots and tubers** | | | | |
| No | 17 (6.0%) | 2 (4.9%) | 19 (5.9%) | 0.774 |
| Yes | 266 (94.0%) | 39 (95.1%) | 305 (94.1%) | |
| **Dark green leafy vegetables** | | | | |
| No | 1 (0.4%) | 0 (0.0%) | 1 (0.3%) | 0.703 |
| Yes | 282 (99.6%) | 41 (100.0%) | 323 (99.7%) | |
| **Vitamin A-rich fruits** | | | | |
| No | 8 (2.8%) | 4 (9.8%) | 12 (3.7%) | 0.028 |
| Yes | 275 (97.2%) | 37 (90.2%) | 312 (96.3%) | |
| **Other vegetables** | | | | |
| No | 65 (23.0%) | 4 (9.8%) | 69 (21.3%) | 0.053 |
| Yes | 218 (77.0%) | 37 (90.2%) | 255 (78.7%) | |
| **Other fruits** | | | | |
| No | 62 (21.9%) | 4 (9.8%) | 66 (20.4%) | 0.071 |
| Yes | 221 (78.1%) | 37 (90.2%) | 258 (79.6%) | |

predominantly starch based such as sorghum, cassava, sweet potatoes, millet and maize. Most of the adolescents in Eastern Uganda reported high consumption of fats/oils and beverages;

**Table 7. Poisson regression of factors associated with undernutrition among pregnant adolescents and young women in Gulu regional referral hospital and St Mary's Hospital Lacor.**

| Variable | | | |
|---|---|---|---|
| | aPR | 95% CI | P-Value |
| **Have you changed from the time you became pregnant?** | | | |
| Yes | Reference | Reference | Reference |
| No | 2.27 | 1.26–4.05 | 0.006 |
| **have you ever been taught about feeding habits during pregnancy?** | | | |
| Yes | Reference | | Reference |
| No | 2.25 | 1.21–4.20 | 0.011 |
| **Milk and milk product** | | | |
| No | Reference | Reference | Reference |
| Yes | 0.44 | 0.24–0.81 | 0.009 |
| **Meat and poultry** | | | |
| No | Reference | Reference | Reference |
| Yes | 0.81 | 0.37–1.78 | 0.604 |
| **Fish and seafood** | | | |
| No | Reference | Reference | Reference |
| Yes | 0.45 | 0.20–1.00 | 0.050 |
| **Eggs** | | | |
| No | Reference | Reference | Reference |
| Yes | 1.01 | 0.47–2.18 | 0.984 |
| **Vitamin A-rich fruits** | | | |
| No | Reference | Reference | Reference |
| Yes | 0.43 | 0.16–1.18 | 0.102 |
| **Other vegetables** | | | |
| No | Reference | Reference | Reference |
| Yes | 4.62 | 1.64–13.01 | 0.004 |
| **Other fruits** | | | |
| No | Reference | Reference | Reference |
| Yes | 0.71 | 0.26–1.93 | 0.505 |

with low intakes of animal source foods, micronutrient source fruits and vegetables.[27]In a study by [28] it was found out major proportion of pregnant women had insufficient minimum dietary diversity intake of 38.8% in district urban general hospital of Ethiopia. According to [29]only one quarter of pregnant women who attended ANC in Shashemane, Oromia region, Central Ethiopia had adequate dietary diversity and 74.6% didn't receive minimum dietary diversity with the least consumed animal products being milk and milk products, fish and eggs. In pregnant women have a low diversity score of 37.1% with very limited number of pregnant women consuming animal source foods like organ meat, fish, eggs, dairy and poultry. In addition, restriction of taking some of the food items due to food taboos such as avoid taking of linseed, honey, milk, and nuts not to give birth of a "fat baby" and avoiding green vegetables not to give birth to a bald baby could be another cause for poor dietary practice of pregnant women in Ethiopia.[30].

The level of under nutrition in this study we carried out was also relatively lower than findings from other different countries, that's to say; 18.1% in Thailand [31], 25% in Bangladesh [32]and 20% in Nigeria [33].The probable reason for the above variation in rates could be little interventions on adolescents' health, nutrition, early marriage, family planning and other adolescent women empowering programs by the government as well as other non-governmental

organizations in the study areas which in turn resulted in chronic energy deficiency, inadequate weight gain during pregnancy and poor nutritional status of adolescent pregnant women. This finding evidenced that, those married before 15 years were more likely undernourished. This was in line with other studies [34]. The practices of unready marriage, which is a case in the study areas, may limit the love, attention and care given by the partner which in turn makes pregnant adolescents work largely on the families' jobs. Adolescent pregnant women having adequate partner, peer or family support were protected from being undernourished. This is in line with study from South Africa [35].

## Strengths and limitations

This study used only MUAC measurement for nutritional assessment which is indirect measure of nutrition and status. The sample size was relatively small, which may limit the generalizability of the results to other populations of PAYW in Uganda. In addition, we conducted a quantitative study and hence did not explore the personal experiences of participants to examine the factors associated with undernutrition. Furthermore, the tools, 24-hour recall and dietary diversity score is not validated in the study area and may not reflect traditional diet in the region.

Major strengths of this study were that the data was collected in the two major highly visited hospital facilities in Gulu City in northern Uganda. We managed to incorporate the women dietary diversity score as a risk factor of undernutrition which was cited as a limitation according to the study by [36]. We recommend further research to be done to assess the qualitative aspect and expound more on the factors associated with undernutrition among pregnant adolescents and young women. Limited studies have been performed in Uganda, we recommend further research in the areas of eastern, western, and central part of Uganda so that we can achieve generalizable prevalence and factors associated with adolescent and young women undernutrition,

The government should sensitize all health workers to ensure that all pregnant mothers especially adolescent mothers are taught about the appropriate diet during pregnancy. Secondly, there should be interventions to focus on nutritional screening for adolescent pregnant women and close monitoring of pregnancy weight gain especially during ANC visits and to deal with the special needs of adolescents to identify malnutrition early and hence prevent the pregnancy related complications. The results of the study will act as a basis for other researchers who would like to carry out similar studies in the study areas or on a similar population. Study results will also ensure policy influence to ensure enhanced nutritional screening and nutritional care among all pregnant women especially those who are adolescents which is in line with the third sustainable development goal 3 (SDG-3) regarding good health and wellbeing for all at all ages. The undernourished mothers in the study were linked to nutritional care.

## Conclusions

In this study about 13% of the participants had undernutrition, translating to about 1 in 8 of PAYW attending GRRH or SMHL had undernutrition, The prevalence of undernutrition was higher among mothers who lacked education about feeding habits during pregnancy and limited access to milk and milk products, fish and seafoods. We recommend interventions focusing on nutritional screening for adolescent pregnant women and close monitoring of pregnancy weight gain especially during ANC visits and to deal with the special needs of adolescents in a friendly manner and health education to pregnant teenagers and their families like avoiding heavy workload while providing adequate support during pregnancy.

## Supporting information

**S1 Dataset.**
(XLSX)

## Acknowledgments

We acknowledge all the study participants for their valuable time and support in completing the study. We also acknowledge administrative support from the study sites.

## Author Contributions

**Conceptualization:** Emmanuel Musinguzi, Peninah Nannono, Moreen Ampumuza, Mathew Kilomero, Brenda Nakitto, Yakobo Nsubuga, Byron Awekonimungu, Rebecca Apio, Moses Komakech, Luke Odongo, Pebalo Francis Pebolo, Felix Bongomin.

**Data curation:** Emmanuel Musinguzi, Peninah Nannono, Moreen Ampumuza, Mathew Kilomero, Brenda Nakitto, Yakobo Nsubuga, Byron Awekonimungu, Rebecca Apio, Moses Komakech, Luke Odongo, Pebalo Francis Pebolo, Felix Bongomin.

**Formal analysis:** Emmanuel Musinguzi, Peninah Nannono, Moreen Ampumuza, Mathew Kilomero, Brenda Nakitto, Yakobo Nsubuga, Byron Awekonimungu, Rebecca Apio, Moses Komakech, Luke Odongo, Pebalo Francis Pebolo, Felix Bongomin.

**Funding acquisition:** Emmanuel Musinguzi, Mathew Kilomero, Brenda Nakitto, Rebecca Apio, Moses Komakech, Luke Odongo, Pebalo Francis Pebolo, Felix Bongomin.

**Investigation:** Emmanuel Musinguzi, Peninah Nannono, Moreen Ampumuza, Brenda Nakitto, Yakobo Nsubuga, Byron Awekonimungu, Rebecca Apio, Moses Komakech, Luke Odongo, Pebalo Francis Pebolo, Felix Bongomin.

**Methodology:** Emmanuel Musinguzi, Peninah Nannono, Moreen Ampumuza, Mathew Kilomero, Brenda Nakitto, Yakobo Nsubuga, Rebecca Apio, Moses Komakech, Luke Odongo, Pebalo Francis Pebolo, Felix Bongomin.

**Project administration:** Emmanuel Musinguzi, Felix Bongomin.

**Resources:** Felix Bongomin.

**Software:** Felix Bongomin.

**Supervision:** Byron Awekonimungu, Moses Komakech, Felix Bongomin.

**Validation:** Peninah Nannono, Moreen Ampumuza, Mathew Kilomero, Brenda Nakitto, Byron Awekonimungu, Rebecca Apio, Moses Komakech, Luke Odongo, Pebalo Francis Pebolo, Felix Bongomin.

**Visualization:** Emmanuel Musinguzi, Peninah Nannono, Moreen Ampumuza, Mathew Kilomero, Brenda Nakitto, Byron Awekonimungu, Rebecca Apio, Moses Komakech, Luke Odongo, Pebalo Francis Pebolo, Felix Bongomin.

**Writing – original draft:** Emmanuel Musinguzi, Peninah Nannono, Moreen Ampumuza, Mathew Kilomero, Brenda Nakitto, Yakobo Nsubuga, Byron Awekonimungu, Rebecca Apio, Moses Komakech, Luke Odongo, Pebalo Francis Pebolo, Felix Bongomin.

**Writing – review & editing:** Emmanuel Musinguzi, Peninah Nannono, Moreen Ampumuza, Mathew Kilomero, Brenda Nakitto, Yakobo Nsubuga, Byron Awekonimungu, Rebecca Apio, Moses Komakech, Luke Odongo, Pebalo Francis Pebolo, Felix Bongomin.

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
