## [Decision Letter · Decision Letter 0]

7 Jul 2024

PONE-D-24-19464Dietary diversity, undernutrition, and predictors among pregnant adolescents and young women attending Gulu University Teaching Hospitals in northern UgandaPLOS ONE

Dear Dr. Bongomin,

Thank you for submitting your manuscript to PLOS ONE. After careful consideration, we feel that it has merit but does not fully meet PLOS ONE’s publication criteria as it currently stands. Therefore, we invite you to submit a revised version of the manuscript that addresses the points raised during the review process.

We look forward to receiving your revised manuscript.

Kind regards,

Mohammed Hasen Badeso, Epidemiologist

Academic Editor

PLOS ONE

Reviewers' comments:

Reviewer's Responses to Questions

**Comments to the Author**

1. Is the manuscript technically sound, and do the data support the conclusions?

Reviewer #1: Yes

Reviewer #2: Yes

2. Has the statistical analysis been performed appropriately and rigorously? 

Reviewer #1: Yes

Reviewer #2: Yes

3. Have the authors made all data underlying the findings in their manuscript fully available?

Reviewer #1: Yes

Reviewer #2: Yes

4. Is the manuscript presented in an intelligible fashion and written in standard English?

Reviewer #1: Yes

Reviewer #2: Yes

5. Review Comments to the Author

Reviewer #1: Dear Authors,

Appreciating the work you have done, please address the following points.

1. Though the manuscript is well written, I have seen some typographical and grammatical errors, and recommend you to go through the entire document. Here I have mentioned some

i. Line 125 "was to be considered normal" - was considered normal

ii. Line 139 "Data analysis plan" - Data analysis

iii. Line 144 "will be described..."- was described

iv. Line 149 -150- "Factors with p<0.05 were considered independent of undernutrition" did you mean "were considered independently/significantly associated with undernutrition"

2. Did you use mean or median for the continuous variables (line 143)? I saw mean age (line 175), mean MUAC (line 186), mean time taken to reach hospital (line 212) etc...

3. It is mentioned that the meantime taken to reach the hospital was 41.784 (line 212). Is it minute?, and is it by walk or car? please mention the means of transport as well.

4. redundancy of ideas- line 141 and 145.

5. The model you used for obtaining dietary diversity information of individual respondents, the 24-h recall method, should be mentioned in the limitations of your study too.

Reviewer #2: The authors present findings from a study estimating the prevalence and risk factors of malnutrition among pregnant adolescent and young women in Northern Uganda.

The article is well written, but requires proofreading prior to publication. For example, "STAT" on line 27 should read STATA.

The mid upper arm circumference and the dietary diversity score both need to be described in more detail/background for readers not familiar with these measures.

The authors present adjusted prevalence rates, but do not specify what was specifically adjusted for and why.

What confounding/prognostic factors were adjusted in regression modeling? For example, age, functional capacity, current health status, etc. Were these factors considered? Please clarify.

In Figure 1, how many participants were excluded due to illness and congenital arm deformities as described in the study population section? These are currently not shown.

In the data analysis plan section, the authors say that regression results were interpreted based on the odds ratio (OR). However, regression model results are provided using the prevalence ratio (PR) in the results section. The odds ratio and the prevalence ratio are distinct measures. Which of these were modeled?

The study does not specify how missing data was managed during the analyses. This information needs to be included.

It is not clear what is meant by "an attempt was made to identify randomized respondents" on line 340. Was randomization employed in this study?

On lines 336-337 the authors state that the cross-sectional nature of the study might affect the establishment of a casual relationship. I find this statement to be misleading and confusing because causation can never be established through an observational cross-sectional study.

On line 279 the authors state "prevalence of undernutrition in our study was small compared...". Should this be "smaller" as a prevalence of 12.7% for undernutrition is not small.

How was antenatal attendance measured/coded? Why does it have 9 levels? This is not clear in Table 1 or described in the text.

6. PLOS authors have the option to publish the peer review history of their article (what does this mean?). If published, this will include your full peer review and any attached files.

Reviewer #1: **Yes: **Semere Welday Kahssay

Reviewer #2: No

---

## [Author Response · Author response to Decision Letter 0]

9 Jul 2024

Dear Dr. Mohammed Hasen Badeso, Epidemiologist

Academic Editor

PLOS ONE

Many thanks for this through work and fast feedback on our manuscript.

We have revised then manuscript according to the reviewers’ comments and we hope this work is now acceptable for publication.

Best,

Dr Felix Bongomin.

PONE-D-24-19464

Dietary diversity, undernutrition, and predictors among pregnant adolescents and young women attending Gulu University Teaching Hospitals in northern Uganda

Reviewer #1: Dear Authors,

Appreciating the work you have done, please address the following points.

1. Though the manuscript is well written, I have seen some typographical and grammatical errors, and recommend you to go through the entire document. Here I have mentioned some

i. Line 125 "was to be considered normal" - was considered normal

ii. Line 139 "Data analysis plan" - Data analysis

iii. Line 144 "will be described..."- was described

iv. Line 149 -150- "Factors with p<0.05 were considered independent of undernutrition" did you mean "were considered independently/significantly associated with undernutrition"

Authors response: Thank you, these have been corrected. 

2. Did you use mean or median for the continuous variables (line 143)? I saw mean age (line 175), mean MUAC (line 186), mean time taken to reach hospital (line 212) etc...

Authors response: This was a typo, we have corrected it to mean and standard deviation.

3. It is mentioned that the meantime taken to reach the hospital was 41.784 (line 212). Is it minute?, and is it by walk or car? please mention the means of transport as well.

Authors response: By walking, tis has been clarified.

4. redundancy of ideas- line 141 and 145.

Authors response: Thank you, we have summarised this and deleted redundant statements.

5. The model you used for obtaining dietary diversity information of individual respondents, the 24-h recall method, should be mentioned in the limitations of your study too.

Authors response: This has been added under strengths and limitations. 

Reviewer #2: The authors present findings from a study estimating the prevalence and risk factors of malnutrition among pregnant adolescent and young women in Northern Uganda.

The article is well written, but requires proofreading prior to publication. For example, "STAT" on line 27 should read STATA.

Authors response: Thank you. This typo has been corrected.

The mid upper arm circumference and the dietary diversity score both need to be described in more detail/background for readers not familiar with these measures.

Authors response: We have described this further. Thank you.

The authors present adjusted prevalence rates, but do not specify what was specifically adjusted for and why.

Authors response: Same as below, we have a sentence on this.

What confounding/prognostic factors were adjusted in regression modeling? For example, age, functional capacity, current health status, etc. Were these factors considered? Please clarify.

Authors response: We have added a sentence on this. Thank you.

In Figure 1, how many participants were excluded due to illness and congenital arm deformities as described in the study population section? These are currently not shown.

Authors response: None had chronic illness or arm anomaly. We have deleted this from the methods section.

In the data analysis plan section, the authors say that regression results were interpreted based on the odds ratio (OR). However, regression model results are provided using the prevalence ratio (PR) in the results section. The odds ratio and the prevalence ratio are distinct measures. Which of these were modeled?

Authors response: We modelled prevalence ratio and used modified Poisson regression.

The study does not specify how missing data was managed during the analyses. This information needs to be included.

Authors response: We have included a sentence in the analysis section.

It is not clear what is meant by "an attempt was made to identify randomized respondents" on line 340. Was randomization employed in this study?

Authors response: Thank you for the guidance, we agree this is confusing and we have deleted it.

On lines 336-337 the authors state that the cross-sectional nature of the study might affect the establishment of a casual relationship. I find this statement to be misleading and confusing because causation can never be established through an observational cross-sectional study.

Authors response: We agree, we have deleted this sentence 

On line 279 the authors state "prevalence of undernutrition in our study was small compared...". Should this be "smaller" as a prevalence of 12.7% for undernutrition is not small.

Authors response: Thank you for this. This has been corrected.

How was antenatal attendance measured/coded? Why does it have 9 levels? This is not clear in Table 1 or described in the text.

Authors response: This was coded as frequency/number of times ANC was attended . We have clarified on this.

---

## [Editor Report · Decision Letter 1]

11 Jul 2024

Dietary diversity, undernutrition, and predictors among pregnant adolescents and young women attending Gulu University Teaching Hospitals in northern Uganda

PONE-D-24-19464R1

Dear Author(s),

We’re pleased to inform you that your manuscript has been judged scientifically suitable for publication and will be formally accepted for publication once it meets all outstanding technical requirements.

Kind regards,

Mohammed Hasen Badeso, Epidemiologist

Academic Editor

PLOS ONE
---

## [Editor Report · Acceptance letter]

15 Jul 2024

PONE-D-24-19464R1 

PLOS ONE

Dear Dr. Bongomin, 

I'm pleased to inform you that your manuscript has been deemed suitable for publication in PLOS ONE. Congratulations! Your manuscript is now being handed over to our production team.

Kind regards, 

on behalf of

Mr Mohammed Hasen Badeso 

Academic Editor

PLOS ONE